# Effects of imatinib on vascular insulin sensitivity and free fatty acid transport in early weight gain

**Camiel V. J. Box**[1][◉], **Amandeep K. Sandhu**[1][◉], **Alexander H. Turaihi**[1], **Pan Xiaoké**[2], **Geesje Dallinga-Thie**[3], **Jurjan Aman**[2‡], **Etto C. Eringa**[1,4‡]*

**1** Department of Physiology, Amsterdam Cardiovascular Sciences (ACS), Amsterdam University Medical Center, Amsterdam, The Netherlands, **2** Department of Pulmonary Medicine, Amsterdam University Medical Center, Amsterdam, The Netherlands, **3** Department of Vascular Medicine, Amsterdam Cardiovascular Sciences (ACS), Amsterdam University Medical Center, Amsterdam, The Netherlands, **4** Department of Physiology, Cardiovascular Research Institute Maastricht (CARIM), Maastricht University, The Netherlands

◉ These authors contributed equally to this work.
‡ These authors also contributed equally to this work.
* e.eringa@amsterdammumc.nl

**Data Availability Statement:** All relevant data are within the manuscript and its Supporting information files.

## Abstract

### Background

Vascular endothelial dysfunction is an essential part of the pathophysiology of type 2 diabetes and its complications. In type 2 diabetes, endothelial dysfunction is characterized by reduced insulin signaling and increased transendothelial transport of fatty acids (FA). As the Abl kinase inhibitor imatinib was previously shown to reverse type 2 diabetes and to inhibit VEGF signaling via Abl kinases, we studied the effect of imatinib on vascular insulin sensitivity and fatty acid transport in vivo and in vitro.

### Methods

C57/BL6J mice were fed a chow diet or Western diet (WD), and received daily imatinib injections for two weeks. Insulin-mediated vasoreactivity of resistance arteries was studied using intravital microscopy, and metabolic insulin sensitivity using the hyperinsulinemic-euglycemic clamp. The effect of imatinib on triglyceride content in skeletal muscle and heart in vivo was also determined. In vitro, the effect of imatinib on fatty acid transport was studied in human umbilical vein endothelial cells (HUVECs) by evaluating the effect of imatinib on fluorescently labeled FA uptake both under basal and VEGF-B-stimulated conditions.

### Results

Imatinib prevented the WD-induced weight gain in mice, independently from food intake. In line with this, imatinib enhanced insulin-mediated vasoreactivity of resistance arteries in the WD-fed mice. However, imatinib did not affect triglyceride content in muscle. In cultured endothelial cells, VEGF-B stimulation resulted in a time-dependent uptake of fatty acids in parallel with increased phosphorylation of the Abl kinase substrate Crk-like protein (CrkL) at

**Funding:** This work was supported by grants from the Netherlands Heart Foundation (ECE, CVON Reconnect and Innovation grants), and the Netherlands Organisation for Scientific Research (ECE, Vidi Grant 917.133.72). The funders had no role in study design, data collection and analysis, decision to publish, or preparation of the manuscript.

**Competing interests:** The authors have declared that no competing interests exist.

Tyr207. Although imatinib effectively prevented VEGF-B-mediated Abl kinase activation, it had no effect on VEGF-B mediated endothelial FA uptake.

## Conclusion

Imatinib prevents weight gain and preserves insulin-mediated vasodilation in WD-fed mice, but does not affect endothelial FA transport despite inhibiting VEGF-B signaling. The beneficial effect of imatinib on insulin-mediated vasodilation may contribute to the anti-diabetic effects of imatinib.

## Introduction

Type 2 diabetes is a progressive metabolic disease that leads to multiple complications, mainly of cardiovascular origin [1]. With an estimated worldwide prevalence of over 300 million patients, type 2 diabetes constitutes a major global health problem [2]. It is characterized by decreased insulin sensitivity and a relative insulin deficiency, resulting in increasing blood glucose levels [3].

Vascular endothelial dysfunction is an essential part of the pathophysiology of type 2 diabetes and its complications [4]. Two hallmarks of endothelial dysfunction observed in type 2 diabetes are vascular insulin resistance, i.e. impairment of vascular insulin signaling [5, 6], and increased transendothelial transport of FFAs [7–9]. Vascular insulin resistance impairs transendothelial transport of insulin to muscle cells [10, 11], as well as insulin-mediated vasodilation [12–15], which in turn determines muscle perfusion. These endothelial functions regulate the access of insulin to myocytes, which is correlated to insulin sensitivity [16], the cornerstone of type 2 diabetes pathophysiology. Importantly, mice fed a high-fat diet to induce insulin resistance showed impaired vascular insulin sensitivity: insulin signaling in endothelial cells was impaired, decreasing muscle perfusion [17].

There is also a prominent increase in free fatty acid (FFA) levels in type 2 diabetes patients [2, 18]. Elevated levels of FFAs increase their transport into organs, such as liver, pancreas, and skeletal muscles, further decreasing the perfusion of these organs and ultimately reducing insulin secretion and insulin sensitivity [18–20]. Increased levels of FFAs directly impair insulin mediated vasodilation [21] and trigger low-grade inflammation, which enhances endothelial permeability [22]. One of the potential regulators of fatty acids uptake is vascular endothelial growth factor-B (VEGF-B), which has been proposed to stimulate transendothelial transport (TET) of lipids [8, 23, 24]. Although regulation of fatty acid transport and blood flow are key features of endothelial dysfunction in type 2 diabetes, few therapeutic agents improve these processes [23]. Since the vascular endothelium plays an important role in the pathophysiology of insulin resistance and type 2 diabetes, it is an attractive target for new therapeutic strategies.

Interestingly, the anti-leukemic drug imatinib has been shown to have glucose-lowering effects: improving fasting blood glucose levels of diabetes patients [25, 26] and insulin sensitivity in high fat diet fed rats [27]. Imatinib is a tyrosine kinase inhibitor that inhibits the kinase activity of among others c-Abl, Abl-related gene (Arg), platelet derived growth factor receptor, c-KIT and discoid domain receptor 1 [19, 25, 26, 28]. In addition to the glucose-lowering effects, imatinib has been shown to improve endothelial barrier dysfunction in mice in vivo through specific inhibition of Abl kinases, and was suggested to increase muscle perfusion [19]. As endothelial barrier dysfunction is an important contributor to type 2 diabetes and

imatinib protects against endothelial dysfunction, imatinib and Abl kinases may regulate endothelial control of fatty acid transport and vascular function in obesity to ameliorate lipotoxicity and impaired glucose uptake.

The aim of our study was to investigate the effect of imatinib on two endothelial functions involved in type 2 diabetes: insulin-stimulated vasodilation of resistance arteries and endothelial fatty acid uptake. We used an in vivo mouse model of early weight gain and lipid accumulation, using mice fed a Western diet (WD) to study the effect of imatinib on metabolic insulin sensitivity, insulin-mediated vasoreactivity of resistance arteries, and triglyceride content in skeletal muscle and the heart [29]. We used a mouse model of early weight gain to specifically study the effect of imatinib on the early pathogenesis of obesity and type 2 diabetes, and endothelial dysfunction is one of the earliest steps in the pathogenesis of type 2 diabetes. In vitro, we used human umbilical vein endothelial cells (HUVECS), to investigate the effect of imatinib on endothelial FA-uptake and studied VEGF-B as a potential stimulus for endothelial fatty acid uptake.

## Methods

### Materials

Vascular endothelial growth factor- B 186 (VEGF-B186) was obtained from R&D systems (Abingdon, Oxfordshire, UK). It was dissolved in phosphate-buffered saline (PBS) to a stock concentration of 30ug/ml [9]. C1-BODIPY®500/510 C12(4,4-Difluoro-5-Methyl-4-Bora-3a,4a-Diaza-s-Indacene-3-Dodecanoic Acid) was from Thermo Fisher Scientific (Waltham, Massachusetts, United States) and dissolved in dimethyl sulfoxide (DMSO) to a stock concentration of 2mM [9]. Imatinib mesylate was from SelleckChem (Houston, TX) and dissolved in dimethyl sulfoxide (DMSO) to a stock concentration of 10mM [20]. Fatty acid-free bovine serum albumin (FF-BSA) was obtained from Sigma-Aldrich (St. Louis, Missouri, United States). The different antibodies used for western blotting i.e. anti-pTyr207CrkL (#3181), anti-CrkL (#3182), phosphor ERK1/2 and total ERK1/2 were from Cell Signalling Technology (Danvers, United States). Human umbilical cords from healthy donors were obtained from Amstelveen Hospital, Amstelveen, Netherlands.

### Animal experimental protocol for imatinib treatment

The investigation conforms to the Guide for the Care and Use of Laboratory Animals published by the US National Institutes of Health (NIH publication No. 85–23, revised 1996), and the work was approved by a national (Central committee for animal experiments; protocol number AVD114002016575) and a local (VU University Amsterdam, Institute for animal welfare (IvD)–Animal experiments; protocol number 575-FYS16-02) ethics committee for animal experiments. The effect of daily imatinib mesylate treatment was evaluated in 12–14-week-old male C57/BL6J mice (Charles River; Amsterdam, the Netherlands) fed either a standard chow diet or a high-fat Western Diet (WD) (D12079B, Research Diets, Inc.). Mice were individually housed for two weeks at a constant temperature of 21±1˚C in a 12h light/12h dark cycle with ad libitum access to water. Mice received daily intraperitoneal injections for a two-week period with either 25 mg/kg imatinib dissolved in 4.0 ml/kg phosphate buffered saline solution (PBS) or only 100 µl PBS as a control. We thus used four different mouse groups: mice fed a chow diet and injected with saline, mice fed a chow diet and injected with imatinib, mice fed a WD and injected with saline, and mice fed a WD and injected with imatinib. General health and behavior of mice were monitored daily. None of the mice died or were euthanized prior to the start of the experiments.

After an overnight fast, mice were initially anesthetized with an intraperitoneal injection of a mixture of fentanyl (0.31 mg/kg), midazolam (6.25 mg/kg), and acepromazine (6.25 mg/kg), and placed on a homeothermic heating pad (Panlab, Vitrolles, France). This specific mixture of anesthesia was chosen, because it maintains normal peripheral insulin sensitivity [30]. The tail vein was cannulated, and a continuous infusion of anesthesia was started (1,3 μg/kg/min fentanyl, 26 μg/kg/min midazolam, and 26 μg/kg/min acepromazine). Then mice underwent the hyperinsulinemic-euglycemic clamp with combined intravital-microscopy (IVM) as described before [20]. The tail vein cannula was used for intravenous infusion of an insulin (Novorapid, Novo Nordisk, France) bolus of 200 mU/kg followed by continuous insulin infusion (7.5 mU/kg/min) for 60 minutes, together with variable infusion of 20% D-glucose to maintain euglycemia at 5.0 mmol/l. In order to measure the diameter of the resistance artery of the gracilis skeletal muscle, we performed intravital microscopy as described [20]. In short, a small incision was made in the skin of the thigh. The femoral vessels with branching muscle resistance artery were exposed and imaged at baseline, after 10, 30, and 60 minutes during the insulin clamp. The insulin-induced-diameter change is the percentage-change of the diameter from baseline.

In 7 of 32 mice, we were not able to calculate the glucose infusion rate (GIR) of the hyperinsulinemic-euglycemic clamp. 6 Mice died unexpectedly during the hyperinsulinemic clamp (most likely caused by over dosage of the anesthesia mixture, which has a small therapeutic window), and one mouse did not reach a steady state of glucose infusion. The other 25 mice did not experience any issues during the clamp. As with the GIR measurements, there are missing data in the resistance artery diameter measurements because of the death of those six mice. Also, technical difficulties with the camera set-up caused unclear images and missing data for the resistance artery diameter in two mice. The missing data were spread out over the different mouse groups.

Mice were sacrificed under full anesthesia by cardiac puncture to extract blood, and organs were subsequently harvested and snap-frozen for determination of muscle triglyceride content.

## Triglyceride content in skeletal muscle, heart and plasma of mice

Lipids were extracted using the NaCl Bligh and Dyer extraction method, as previously described [31]. The method in brief: 100 mg tissue (skeletal muscle/heart) was homogenized in 400 μl cold PBS in the magna lyzer tubes. Then, 1.5 ml methanol/chloroform (2:1) was added, followed by centrifugation (2000 rpm, 5 min at 4˚C). Subsequently, 0.5 ml chloroform and 0.5 ml NaCl was added to the supernatant, followed by centrifugation. After centrifugation, the chloroform layer was dried under nitrogen, and then the pellet was re-suspended in 2% triton-X100. After that, the triglycerides were measured with the enzymatic-colorimetric method (GPO-PAP method).

## HUVEC culture

Human umbilical vein endothelial cells (HUVECS) were isolated from human umbilical cords of healthy donors and cultured in medium M199 at 37˚C and 5% CO2, as previously described [19]. The culture medium was changed on alternate days. The composition of the culture medium was as follows: M199 medium (Biowhittaker/Lonza), penicillin (100 U/ml), strepto-mycin (100 μg/ml) (Biowhittaker/Lonza), heat-inactivated human serum 10% (Sanquin Blood Supply, Amsterdam, The Netherlands), heat-inactivated new-born calf serum 10% (Gibco, Grand Island, NY), crude endothelial cell growth factor 150 μg/ml (prepared from bovine brains), L-glutamine 2 mmol/L (Biowhittaker/Lonza) and heparin 5 U/ml (Leo

Pharmaceuticals products, Weesp, The Netherlands). The cells between passages 1 and 2 were used for the experiments. All experiments were repeated with cells from three different donors.

## Western blot

For Western blot analysis, cells were seeded as described above. The cells were pre-incubated in a medium without growth factors (1% FF-BSA, plain M199) containing inhibitors or vehicle as indicated, and stimulated with growth factor or vehicle for indicated intervals. After stimulation, cells were lysed, and 36 μg of protein was electrophoresed in 8% acrylamide gels and transferred to PVDF membrane, as previously described [19]. Phosphorylation status and protein expression were determined by overnight (4˚C) incubation of the PVDF membrane with indicated antibodies. The following antibodies were used: phospho-ERK1/2 and total ERK1/2 (dilution of 1:1000), pTyr207CrkL and total CrkL (dilution of 1:500). Using HRP-labelled secondary antibodies, chemoluminiscence was used for detection.

**Real time quantitative polymerase chain reaction (qRT-PCR).**   Total RNA was extracted from Gastrocnemius muscle using a miRCURY RNA isolation kit (Exiqon). Muscle RNA was reversely transcribed and amplified using Ovation PicoSL WTA System V2 (Nugen). Quantitative PCR was performed using a commercial SYBR green mastermix (Biorad) and specific primers for VEGF-B as described by Hagberg et al. [23] and for VEGFR1, IRS1 and Glut4. Primers for IRS1 were described in Meijer et al. [29], primers for VEGFR1 were described in Robciuc et al. [32]. Data were corrected for the geometric mean of Rps15 and expressed relative to the muscle of vehicle-treated, chow-fed mice.

## BODIPY-FA uptake assay

HUVECS were seeded (1:1.9 density) on Ibidi arrays (1cm$^2$) coated with 1% gelatin, followed by crosslinking with 0.5% glutaraldehyde, and grown to confluence with change of culture medium every other day. For stimulation, cells were incubated with 1% FF-BSA in plain M199 with inhibitors or vehicle, followed by stimulation with growth factors at indicated concentrations and intervals. The fluorescently labelled fatty acids (BODIPY-FA), C1-BODIPY®500/510 C12 (20 μM) were added (3 minutes at 37˚C) after washing the cells with 1% FF-BSA, PBS. Subsequently, the cells were fixed with 4% paraformaldehyde (PFA), as previously described [8]. They were imaged with 40x objective and equal exposure times.

## Statistics

Data are expressed as mean ± standard deviation. To assess the effects of the Western diet and chronic imatinib treatment, two-way ANOVA tests were used to test the differences in means between the four mouse groups: SalineChow, ImatChow, SalineWD, and ImatWD. To test the difference between baseline and intervention (hyperinsulinemia) values Wilcoxon matched-pairs signed rank tests were used. P-values of <0.05 were considered significant. We used a power calculation to determine the minimal number of mice needed to be able to get statistically significant results. We calculated that we would need at least 7 mice in each group, added 10% for any unforeseen complications and thus we determined to use groups of 8 mice. The calculation was as follows: $z_{\alpha/2}$—$z_\pi$ = $(n/2)^{1/2}$ * $(\mu_1 - \mu_2)/\sigma$, with $z_{\alpha/2}$-$z_\pi$ = 3,242 (α = 5% and π = 90%), $(\mu_1 - \mu_2)$ = (2,0–1,3) = 0,7 and σ = 0,4. This leads to n = 7.

The effect of imatinib on triglyceride content in skeletal muscle, heart and plasma of mice was analyzed using ordinary two-way ANOVA. The effect of VEGF-B on CrkL and ERK phosphorylation was analyzed using repeated measures one-way ANOVA. A two-way repeated measures ANOVA was used to analyze the effect of imatinib on CrkL and ERK

phosphorylation. The effect of VEGF-B concentration and time point on the uptake of fatty acids was measured using ordinary two-way ANOVA. The effect of imatinib on the uptake of fatty acids was assessed using ordinary one-way ANOVA. N represents the number of times an experiment was repeated.

## Results

### Imatinib protects against Western diet-induced weight gain in mice, without affecting metabolic insulin sensitivity or glucose metabolism

During the two-week diet and treatment intervention we measured food intake and body weight of the mice. We observed a similar intake ($P = 0.08$) in imatinib treated mice (39.8±2.6 g by ImatChow, and 33.5±3.9 g by ImatWD mice) as in saline treated mice (41.7±1.7 g by SalineChow and 36.0±4.4 g by SalineWD mice). The body weight of WD-fed mice (25.4±3.4 g for SalineWD and 25.2±3.3 g for ImatWD) was significantly higher ($P = 0.007$) than the body weight of chow-fed mice (23.2±1.5 g for SalineChow and 22.0±1.6 g for ImatChow) (Fig 1A).

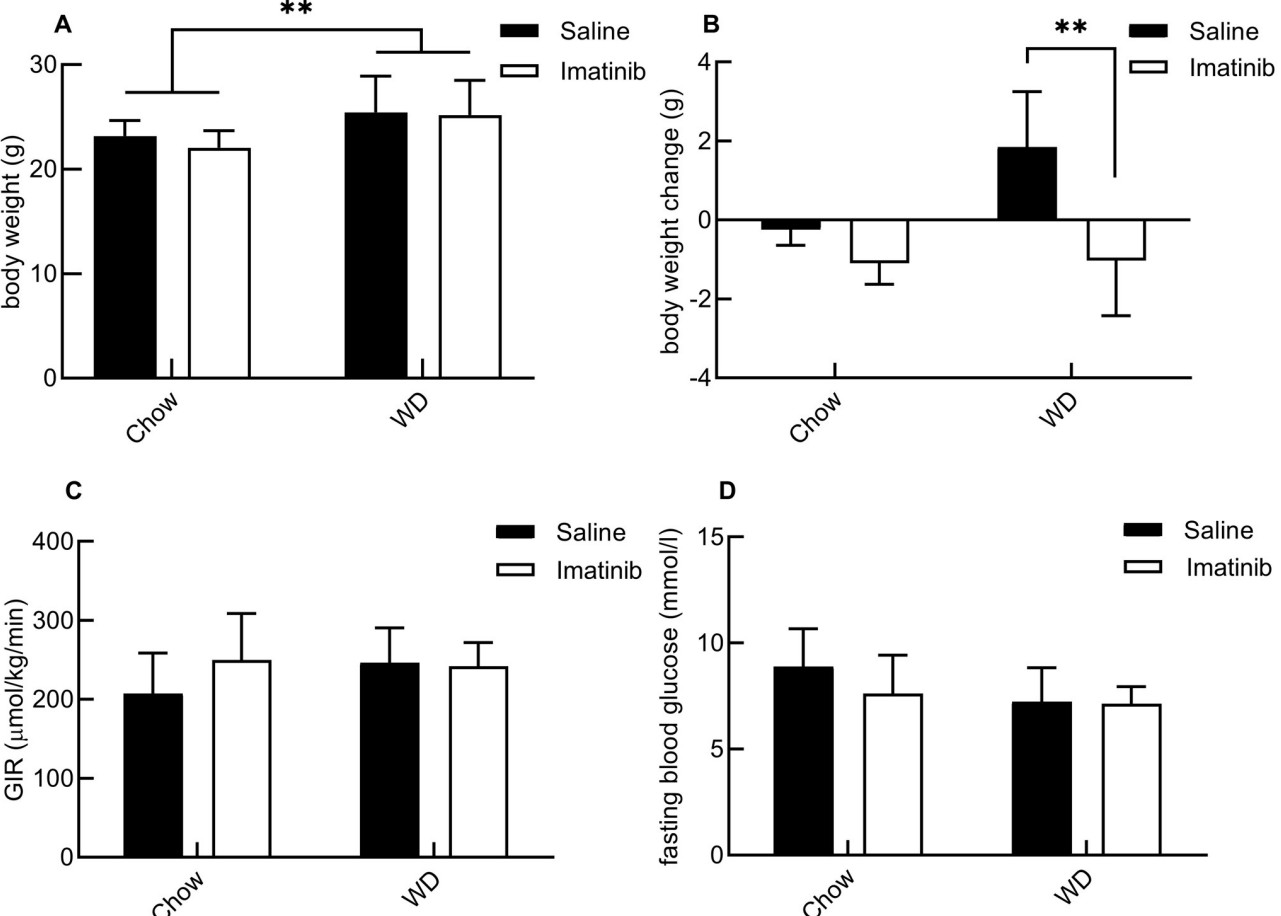

**Fig 1. The effect of imatinib and Western Diet (WD) on body weight, metabolic insulin sensitivity and glucose metabolism.** A: WD-fed mice had higher body weight than chow fed mice after two weeks of diet (P = 0.007, n = 8). B: Two-week WD feeding caused bodyweight gain in mice given saline injections. This weight-gaining effect was rescued by imatinib treatment (P = 0.006, n = 8). C: Glucose infusion rate (GIR) quantifies whole-body glucose uptake during the hyperinsulinemic-euglycemic clamp. Two-week Western diet feeding and imatinib treatment did not affect GIR: P = 0.42 and P = 0.32, respectively (n = 5–7). D: Two-week Western diet feeding and imatinib treatment did not affect fasting glucose levels: P = 0.07 and P = 0.23, respectively (n = 7–8). Results were tested for significance with two-way ANOVA tests. ***P<0.01.

More interestingly, WD feeding caused an increase of 1.85±1.4g in body weight in the saline-treated mice, but not in imatinib-treated mice ($P = 0.006$). Instead, a weight loss of -1.03±1.4 g was observed in imatinib-treated mice that were fed a WD (Fig 1B). These data show that imatinib protects against Western diet-induced weight gain, without significantly affecting food intake. To investigate the effects of imatinib on metabolic insulin sensitivity, we quantified insulin-induced whole-body glucose uptake as the glucose infusion rate (GIR) during the hyperinsulinemic-euglycemic clamp in mice fed either standard chow diet or WD. We found no effect of the WD or imatinib treatment on insulin-induced whole-body glucose uptake. The GIRs after two weeks of treatment and diet were comparable between the groups: (in μmol/kg/min) 207.5±51.0 SalineChow, 250.0±58.7 ImatChow, 246.4±44.0 SalineWD, and 241.7±30.1 ImatWD (Fig 2a). So, imatinib ($P = 0.32$) and WD-feeding ($P = 0.42$) did not affect metabolic insulin sensitivity in our study. Also, fasting blood glucose levels as measured at the start of the clamp were similar in the mouse groups: (in mmol/l) 8.9±1.8 SalineChow, 7.6±1.8 ImatChow, 7.2±1.6 SalineWD, and 7.1±0.8 ImatWD (Fig 2b). This shows that glucose metabolism was not significantly affected by the WD ($P = 0.07$) or imatinib treatment ($P = 0.23$). This is in line with the results that imatinib and WD did not influence the GIR and together these data show that two-week WD and imatinib treatment did not affect metabolic insulin sensitivity or glucose metabolism in these mice.

## Imatinib preserves insulin-induced vasodilation of muscle resistance arteries in WD-fed mice in vivo

By surgically exposing the gracilis artery before starting the hyperinsulinemic-euglycemic clamp, we were able to evaluate the vasodilating effect of insulin in vivo. While imatinib did not affect the vasodilator effect of insulin in chow-fed mice (Fig 2a), it did preserve this capacity in WD-fed mice. In WD-fed mice treated with saline we did not observe any vasodilation during hyperinsulinemia, but in the WD fed mice treated with imatinib we did observe vasodilation during hyperinsulinemia: in the SalineWD group the relative diameter (in percentage-change from baseline) decreases from 11.1±6.6% at 30 min to 5.9±8.3% at 60 min, while in the ImatWD group the relative diameter kept increasing from 13.8±11.0% at 30 min to 19.6±8.4% at 60 min. This is a significant difference ($P = 0.03$) in diameter change at t = 60 min between the SalineWD and the ImatWD groups (Fig 2b).

These data show that a two week treatment with imatinib prevents the loss of insulin-induced vasodilation that is observed after a Western diet.

## Effect of imatinib on triglyceride content (TG) in skeletal muscle, heart, and plasma of mice

As imatinib protected mice against Western diet-induced weight gain, we investigated the effect of imatinib on triglyceride content in skeletal muscles, heart, and plasma of mice fed with either chow or WD. Western diet increased TG content in skeletal muscle ($P = 0.009$) compared to chow diet, but this effect of Western diet was not seen in the heart ($P = 0.1$). Imatinib did not affect TG content in skeletal muscle in WD- ($P = 0.5$) or chow-fed mice (P>0.9) (Fig 3A). Similarly, no effect of imatinib was observed on TG content in the heart (chow versus WD group P>0.9), and plasma (chow versus WD group P>0.9) (Fig 3B and 3C).

As VEGF-B was shown to mediate the effects of a high-fat diet on triglyceride accumulation in muscle [23], we examined VEGF-B expression in gastrocnemius muscles of mice. All mice showed robust expression of VEGF-B in muscle, but neither the WD nor imatinib altered VEGF-B expression in muscle (Fig 3D). In contrast, WD increased the expression of VEGFR1 by ±30% in muscle, and imatinib did not alter this effect (Fig 3E). WD also increased the

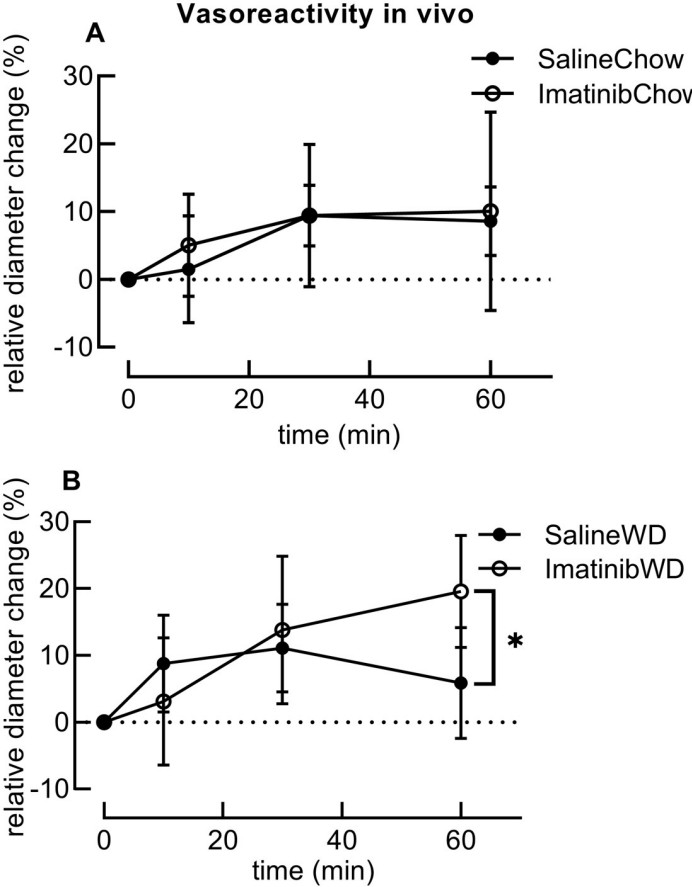

**Fig 2. Vasoreactivity of the gracilis artery in response to insulin, measured with intravital microscopy.** A: Imatinib does not affect insulin-induced vasoreactivity in vivo in mice fed a chow diet. B: Imatinib increases insulin-induced diameter change at t = 60 minutes in mice fed a Western diet (WD). $^*P<0.05$ as tested with unpaired T-test; n = 4–8 per group.

expression of Glut4 by ±2-fold (Fig 3G), but not of the insulin receptor substrate 1 (Fig 3F). Imatinib did not change the gene expression of either Glut4 or IRS1 (Fig 3F and 3G).

These data show that a short term WD induced triglyceride accumulation in skeletal muscle and increased expression of VEGFR1, without upregulation of VEGF-B. Imatinib did not decrease triglyceride content in skeletal muscle, heart, or plasma of Western diet-fed mice and did not alter the effects of WD on gene expression in muscle, despite its protective effect on weight gain.

## Effect of imatinib on endothelial FA uptake and on phosphorylation of CrkL in vitro

As imatinib prevented Western diet induced weight gain in the in vivo experiments but did not affect the uptake of triglycerides in vivo, in vitro experiments were performed to validate the effect of imatinib on fatty acid uptake in human endothelial cells. As VEGF-B reportedly enhances fatty acid transport and triglyceride accumulation was accompanied by increased VEGFR1 expression in muscle, we specifically evaluated VEGF-B mediated fatty acid uptake.

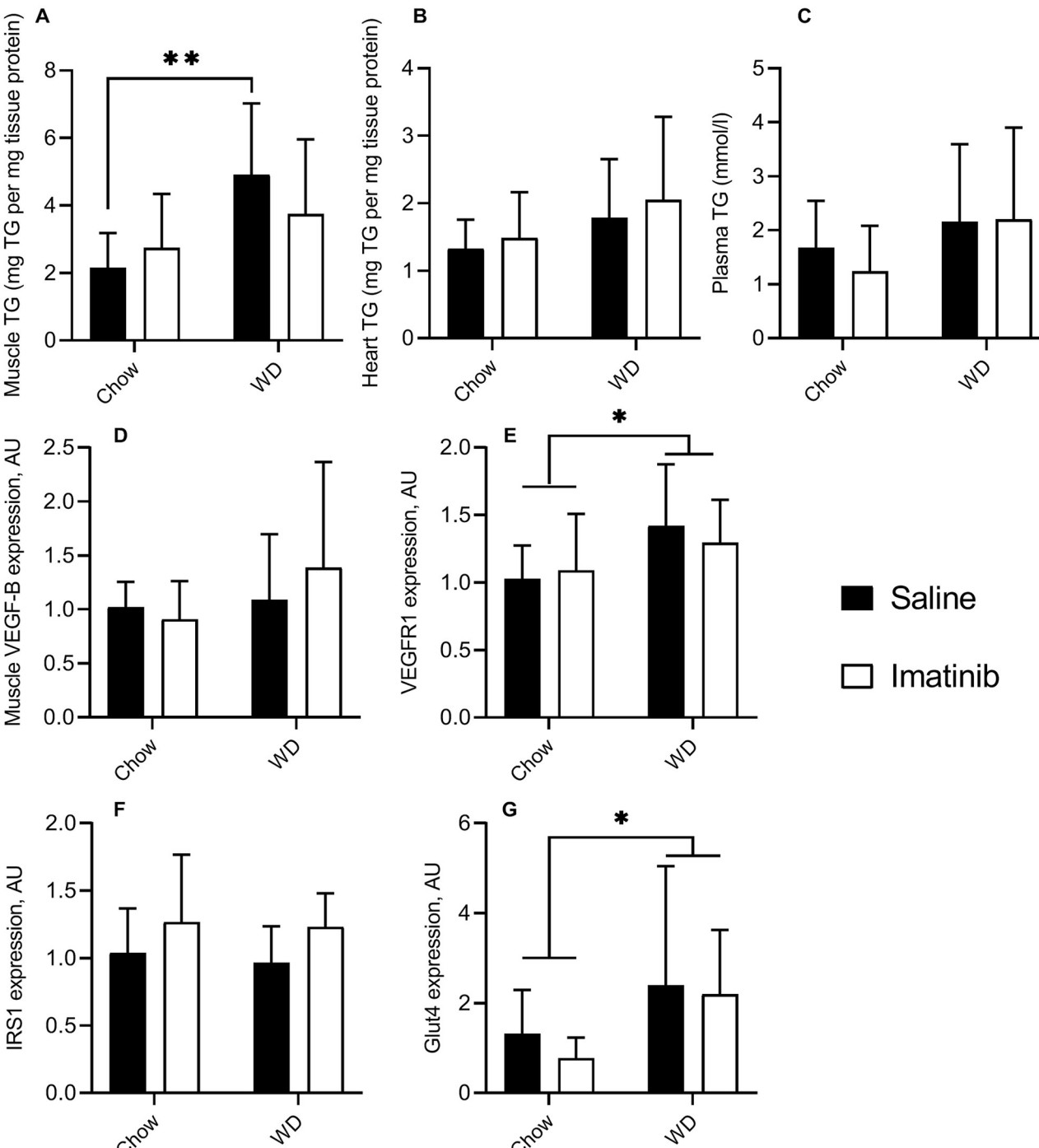

**Fig 3. Effect of WD and imatinib on triglyceride (TG) content of skeletal muscle, heart, and plasma and on muscle gene expression.** A: TG content in skeletal muscle: WD increased TG content in skeletal muscle as compared to chow ($P = 0.009$). Imatinib did not decrease TG content in the chow group ($P>0.9$) and in the WD group ($P = 0.5$). B: TG content in heart: WD caused no increase in TG content as compared to chow diet ($P = 0.1$). Imatinib caused no effect on TG content. C: TG content in plasma: imatinib did not decrease the TG content in the plasma of WD fed mice. D: Expression of VEGF-B in gastrocnemeus muscle: imatinib ($P = 0.27$) and WD ($P = 0.7$) did not affect VEGF-B expression. E: Expression of VEGFR1 in gastrocnemeus muscle: WD feeding increased VEGFR1 expression compared to Chow feeding ($P = 0.04$), while imatinib did not affect the expression ($P = 0.5$). F: Expression of IRS1 in gastrocnemeus muscle: imatinib ($P = 0.08$) and WD ($P = 0.7$) did not affect IRS1 expression. G: Expression of Glut4 in gastrocnemeus muscle: WD feeding increased Glut4 expression compared to Chow feeding ($P = 0.04$), imatinib had no effect on the expression ($P = 0.5$). $^{*}P<0.05$ and $^{**}P<0.01$ in Bonferroni post hoc test of non-repeated measures ANOVA, a.u. = arbitrary units. N = 8 (number of mice in each group).

## VEGF-B enhances the phosphorylation of CrkL in HUVECS and imatinib inhibits the VEGF-B mediated phosphorylation of CrkL

After evaluating the optimal concentration and time point of VEGF-B, we determined the effect of VEGF-B on the phosphorylation of CrkL, a downstream target of Abl kinases. The treatment of HUVECS with VEGF-B (100 ng/ml) caused a time-dependent increase in the phosphorylation of CrkL ($P = 0.019$ at 15 minutes of VEGF-B stimulation) (Fig 4A and 4C). To elucidate the effect of imatinib on VEGF-B-mediated CrkL phosphorylation, HUVECS were pre-incubated 1 hour with imatinib (10 μM) or DMSO (0.1%), followed by addition of VEGF-B at different time points: 5 min, 15 min, and 30 min. Imatinib caused a significant decrease in VEGF-B-mediated phosphorylation of CrkL as compared to DMSO at 15 min ($P = 0.005$) and 30 min ($P = 0.019$) stimulation with VEGF-B. However, imatinib did not decrease the CrkL phosphorylation in non-stimulated conditions or in conditions with 5 min VEGF-B stimulation (Fig 4B and 4D). To demonstrate the selectivity of imatinib for CrkL phosphorylation and Abl kinase activity, we evaluated the effect of imatinib on ERK1/2 phosphorylation, a pathway that is activated by VEGF but independent from Abl kinases. VEGF-B

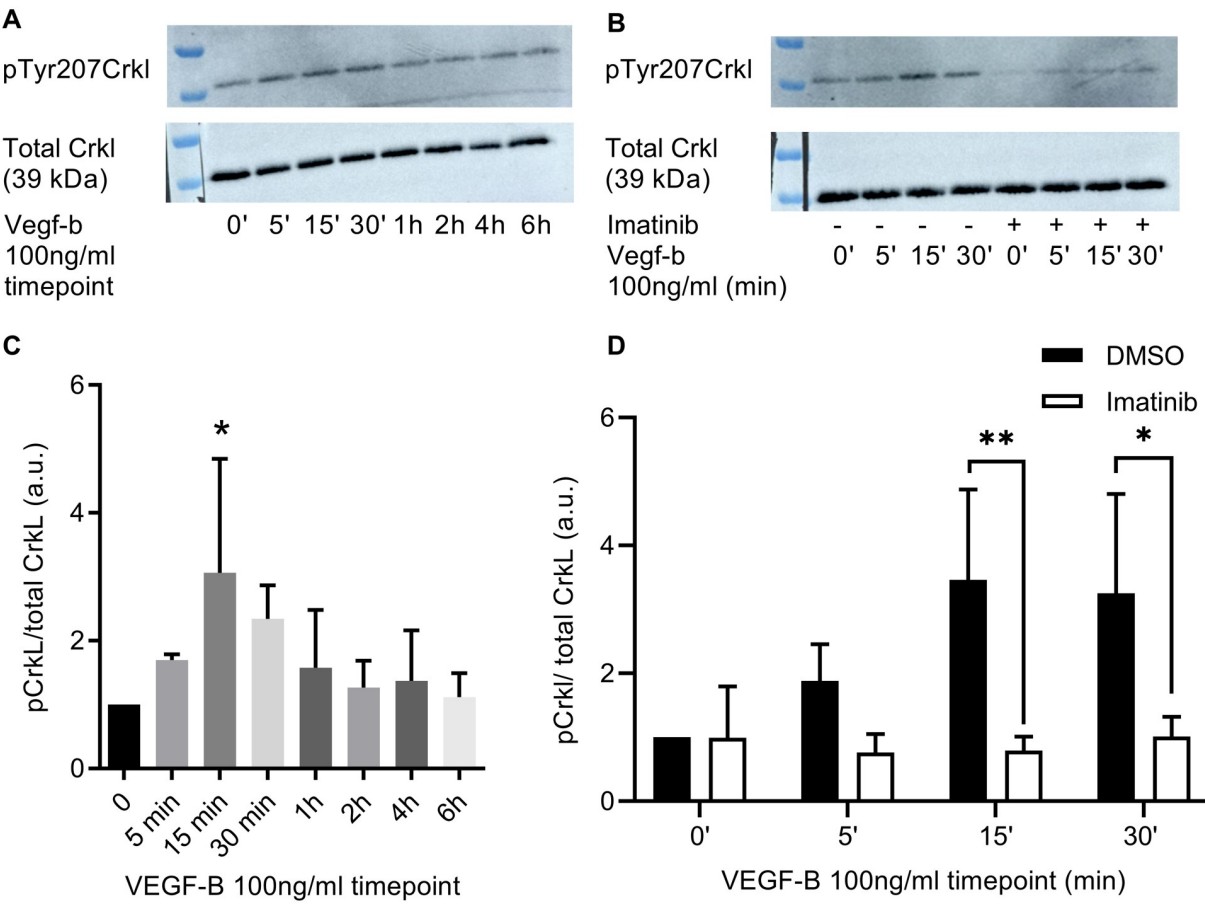

**Fig 4. Effect of vascular endothelial growth factor-B (VEGF-B) and imatinib on Crk-like protein (CrkL) phosphorylation.** A: Western blot images of VEGF-B-mediated phosphorylation of CrkL at different time points. C: Quantification of A. The phosphorylation of CrkL showed an increase at 15 min stimulation with VEGF-B. *$P = 0.019$ in Bonferroni post hoc test of repeated measures one-way ANOVA. B: Western blot images of the effect of imatinib on VEGF-B-mediated CrkL phosphorylation. D: Quantification of B. Imatinib attenuates the phosphorylation of CrkL in conditions with 15 and 30 min stimulation with VEGF-B. **$P < 0.01$, *$P < 0.05$ in Bonferroni post hoc test of repeated measures two-way ANOVA. a.u. = arbitrary units; n = 3. For uncropped blot images see S1 Fig.

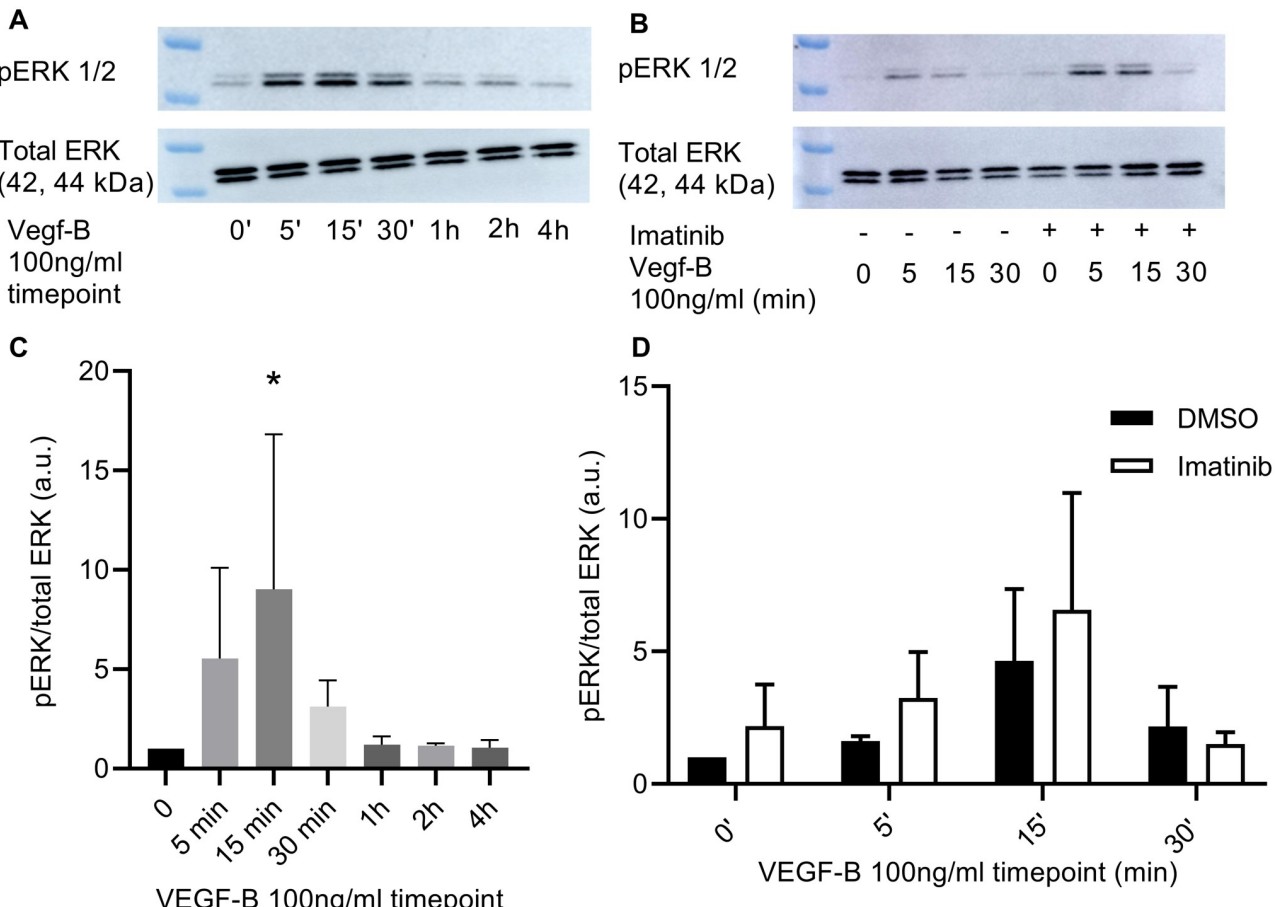

**Fig 5. Effect of vascular endothelial growth factor-B (VEGF-B) and imatinib on ERK1/2 phosphorylation.** A: Western blot images of VEGF-B mediated phosphorylation of ERK1/2 at different time points. C: Quantification of A. The treatment of HUVECS with VEGF-B enhanced the phosphorylation of ERK1/2 at 15 min stimulation with VEGF-B. *$P = 0.04$ in Bonferroni post hoc test of repeated measures one-way ANOVA. B: Western blot images of the effect of imatinib on VEGF-B mediated ERK1/2 phosphorylation ("-" = DMSO; "+" = Imatinib). D: Quantification of B. Imatinib caused no effect on the phosphorylation of ERK1/2 in basal conditions and in conditions treated with VEGF-B. *$P<0.05$ in Bonferroni post hoc test of repeated measures two-way ANOVA. a.u. = arbitrary units; n = 3. For uncropped blot images see S1 Fig.

increased ERK1/2 phosphorylation in a time-dependent manner ($P = 0.04$ at 15 min of VEGF-B stimulation), which was not affected by pretreatment with imatinib ($P>0.9$) (Fig 5A–5D).

Taken together, these data show that VEGF-B increases Abl kinase activity, a process that is specifically targeted by the Abl kinase inhibitor imatinib.

## VEGF-B increases fatty acid uptake in HUVECS independent of Abl kinases

To determine the effect of VEGF-B on endothelial transport of fatty acids, we measured the fluorescence as a measure of BODIPY-FA (C1-BODIPY®500/510 C12 (20 μM)) uptake. The treatment of HUVECS with VEGF-B 100 ng/ml and VEGF-B 300 ng/ml resulted in a time-dependent increase in the uptake of BODIPY-FA at 2 hours ($P = 0.02$ and $P = 0.02$) and 6 hours ($P = 0.01$ and $P = 0.03$) of stimulation with VEGF-B (Fig 6A and 6B). To evaluate the effect of imatinib on the uptake of fluorescently labelled fatty acids, we treated the cells with

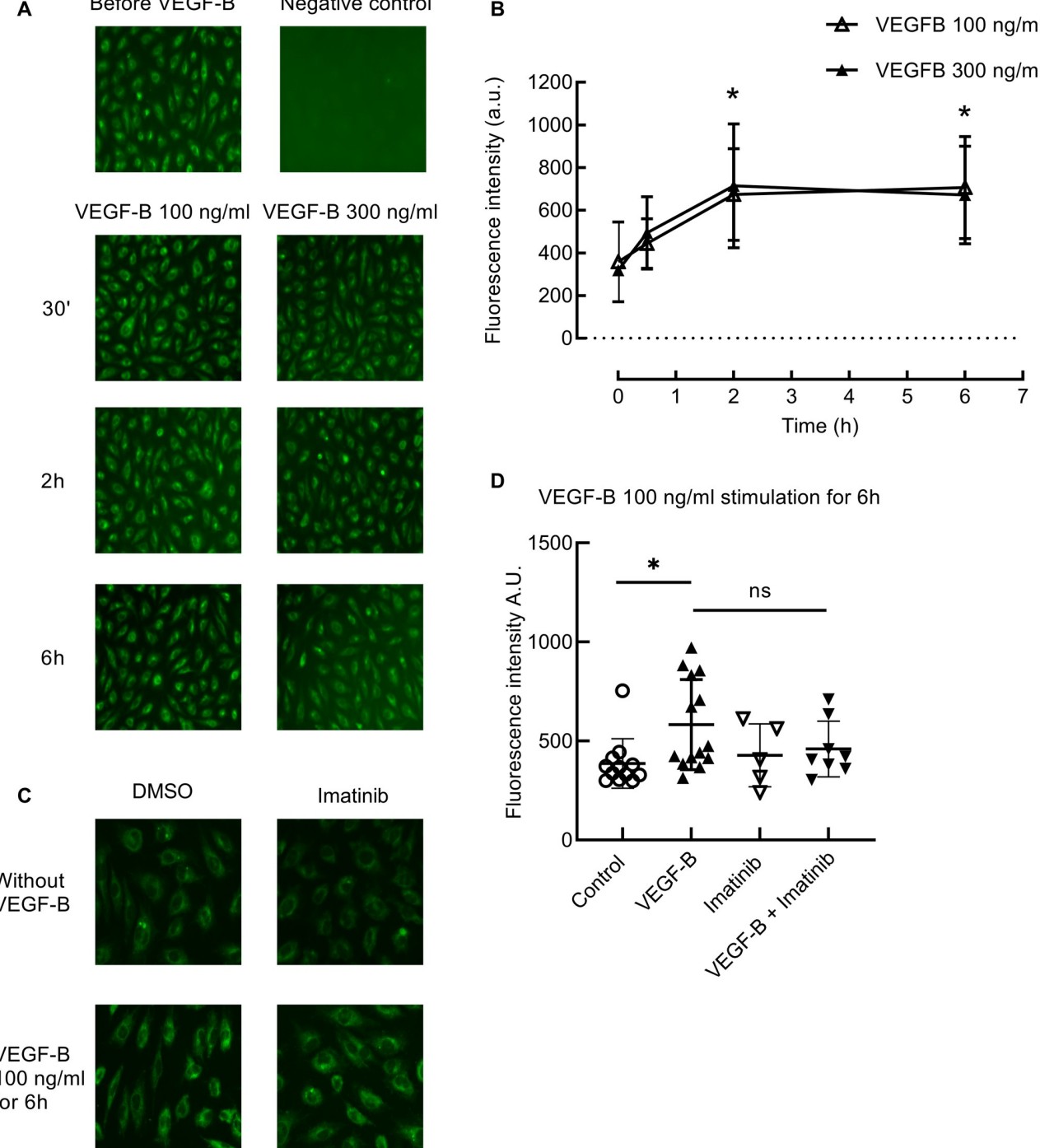

**Fig 6. Effect of VEGF-B and imatinib on Bodipy-FA uptake into HUVECS.** A: Bodipy-FA uptake in cells treated with VEGF-B 100 ng/ml and VEGF-B 300 ng/ml at different time points. B: Quantification of fluorescence in A. VEGF-B 100 ng/ml and 300 ng/ml caused a time-dependent increase in the uptake of Bodipy-FA at 2 hours ($P = 0.02$ and $P = 0.02$) and 6 hours ($P = 0.01$ and $P = 0.03$) of VEGF-B stimulation. *$P$ in ordinary two-way ANOVA. a.u. = arbitrary units; n = 6. The data in this experiment are partly the same as the data in experiment Fig C, D. C: BODIPY-FA uptake into HUVECS treated with imatinib, with or without 6h 100 ng/ml VEGF-B stimulation. D: Quantification of fluorescence in B. Six hours stimulation with 100ng/ml VEGF-B caused an increase in the uptake of fatty acids in HUVECS. *$P = 0.02$ in ordinary one-way ANOVA. However, imatinib caused no decrease in Bodipy-FA uptake under basal and VEGF-B treated conditions. *$P = 0.96$ and $P = 0.32$ in ordinary one-way ANOVA. Non-significant (ns). N = 5–14.

either imatinib 10 μM or DMSO 0,1% and stimulated them with or without VEGF-B 100 ng/ml for 6 hours. Imatinib did not decrease the uptake of BODIPY-FA in the cells treated with VEGF-B (*P* = 0.32) or under basal condition (*P* = 0.96) (Fig 6C and 6D), in line with our in vivo data showing that imatinib did not affect muscle triglyceride content.

Together these in vitro data show that VEGF-B stimulation caused a time-dependent increase in the uptake of fatty acids corresponding with increased phosphorylation of Arg substrate CrkL at Tyr207. Although imatinib inhibited the VEGF-B mediated CrkL phosphorylation, it did not affect endothelial FA uptake.

## Discussion

This study demonstrates that imatinib prevents Western diet-induced weight gain and preserves insulin-induced vasodilation of resistance arteries in WD-fed mice. In vitro, we found that VEGF-B increased the uptake of fatty acids in a time-dependent manner. Although imatinib inhibited VEGF-B mediated signaling, it did not change FA uptake. Insulin-induced vasodilation of resistance arteries, i.e. vascular insulin sensitivity, and fatty acid uptake are two interrelated endothelial functions that contribute to the pathophysiology of type 2 diabetes. Here, we show that imatinib protects against WD-induced weight gain most likely by improving insulin-induced vasodilation of resistance arteries, without affecting the uptake of fatty acids.

### Imatinib protects insulin-induced vasodilation during a Western diet independently from insulin-dependent glucose uptake

Our in vivo mouse model included mice fed a Western diet as a model for early diet-induced weight gain and insulin resistance. High-fat diet feeding causes vascular and metabolic insulin resistance and inflammation as well as an increase in plasma lipid levels and lipid deposition in organs [22, 33, 34]. Remarkably, vascular insulin resistance and inflammation precede metabolic insulin resistance [22, 34], underscoring the importance of endothelial dysfunction in the pathophysiology of type 2 diabetes. A human study from our group showed that some of the earliest effects of a high-fat diet on endothelial function were seen in resistance arteries of skeletal muscles, where insulin-induced vasodilation was decreased while whole-body glucose uptake was not yet affected [35]. Here, we find the same early effects of a high-fat diet in mice. We also show here that imatinib protects against weight gain, most likely via enhancing insulin-mediated vasodilation of resistance arteries. Since imatinib preserves this important characteristic of normal vascular insulin sensitivity, and vascular insulin signaling plays an important role in the pathophysiology of type 2 diabetes and its complications [36], the antidiabetic effect of imatinib might well be caused by its protective effect on this endothelial function. Future studies are required to reveal the molecular mechanisms that underlie the effect of imatinib on insulin-mediated vasodilation, and to confirm the link between insulin-mediated vasodilation and weight gain that is suggested by our study.

In contrast to results in other studies [22, 29], we did not observe changes in GIR after two weeks of WD, meaning metabolic insulin sensitivity was not decreased, while vascular insulin sensitivity was decreased. Since we used a model for early weight gain, this suggests, as other research did before, that vascular insulin resistance precedes metabolic insulin resistance in the pathogenesis of type 2 diabetes [22]. Despite weight gain and muscle triglyceride accumulation, we did not observe whole-body insulin resistance in this short-term Western diet treatment. This result was corroborated by a lack of differences in IRS1 expression and even an increase in muscle Glut4 expression (Fig 3F and 3G). This lack of insulin resistance despite lipid accumulation in muscle agrees with findings in humans [37] and suggests that these two

are separate phenomena, at least during short term lipid accumulation, and that impairment of insulin-induced vasodilation alone is not sufficient to induce insulin resistance. Despite the lack of an effect of the WD on whole-body glucose uptake, we show here that imatinib protects insulin-induced vasodilation, a measure for vascular insulin sensitivity, in mice fed a WD independently of glucose uptake.

## Imatinib prevents weight gain by Western diet feeding independently from plasma and tissue triglycerides and VEGF-B

Western diet feeding increases body-weight by inducing accumulation of lipids and fat tissue. Since we found here that imatinib protects against weight-gain by WD feeding, we hypothesized that imatinib affects lipid metabolism. However, both the in vivo and in vitro experiments presented in our study failed to support this hypothesis as no effect of imatinib on lipid uptake or content was found. Importantly, our data about weight gain are consistent with other studies: imatinib has been shown to normalize triglyceridemia in diabetes patients [31], to decrease liver steatosis and hepatic triglyceride concentrations [38, 39], and to decrease body weight and body fat [27]. One study in humans showed an imatinib-induced increase in adiponectin levels in non-diabetic patients treated with imatinib [40]. Interestingly, adiponectin, a hormone secreted by adipocytes, regulates insulin sensitivity as well as insulin-induced vasodilation [41, 42]. However, we did not find an effect of imatinib on triglyceride content in skeletal muscle, heart, or plasma. The reason for this lack of effect of imatinib on TG content could be the shorter duration of imatinib treatment used in our study (2 weeks), as compared to for example the study of Han et al, in which mice were treated with imatinib for 4 weeks [38]. Alternatively, it could simply mean that imatinib does not exert its anti-diabetic effect through lipid metabolism. Although a definite answer remains to be provided, our data do support a role of imatinib in weight regulation independent of triglycerides and fatty acids.

## Abl kinases mediate VEGF-B signaling without affecting VEGF-B-mediated uptake of fatty acids

VEGF-B has been reported to enhance cellular uptake of fatty acids in animal models of diabetes and mouse bEnd3 endothelial cells [8, 23, 43], and one study reported increased levels of circulating VEGF-B in obese patients [44]. In line with these studies, we found that VEGF-B causes a time-dependent increase in the uptake of fatty acids in HUVECS. However, this is in contrast with the study conducted by Robciuc et al. [32], who found that enhanced VEGF-B/VEGR1 signaling improves glucose metabolism in obese mice, through increased VEGF-A activity, angiogenesis and thermogenesis in adipose tissue. Our in vitro data support a direct effect of VEGF-B on fatty acid uptake in endothelial cells. This direct effect of VEGF-B suggests involvement of VEGFR1, since VEGF-B is known to bind specifically to this receptor. However, the role of VEGF-B in fatty acid transport and its enhancing effect on VEGF-A have been debated. Seki et al. [45] found that in a diet-induced obesity mouse model, endothelial deletion of VEGFR1 promotes white adipose tissue browning, decreases fatty liver and improves insulin sensitivity. In this paper, they proposed that VEGFR1 acts as a negative regulator for VEGF-A [45], instead of a positive regulator as proposed by Robciuc et al. [32]. Our data support that VEGF-B directly stimulates fatty acid transport, independently from VEGF-A, likely via VEGFR1. The normal VEGF-B and increased VEGFR1 expression in muscles of Western diet-fed mice suggests that lipid accumulation in muscle is mediated by increased VEGFR1 signaling rather than increased VEGF-B expression.

High serum levels of free fatty acids activate inflammatory signaling in the endothelium, which can compromise endothelial insulin signaling [21] and barrier function [46]. We found

that VEGF-B enhances the phosphorylation of CrkL, a downstream target of Abl kinases c-Abl and Arg. As VEGF-B is known to activate only VEGFR1, this suggests that VEGF-B binding to VEGR1 activates Abl kinases. Although VEGF-B increased phosphorylation of CrkL (downstream target of Arg/c-Abl kinases), no effect of imatinib was observed on VEGF-B induced phosphorylation of ERK. This indicates that ERK1/2 and Abl kinase are independently activated by binding of VEGF-B to VEGFR1, and that imatinib selectively inhibits Abl kinase activation.

Our observations confirm that VEGF-B induces cellular FA uptake, that this effect is independent from Abl kinase and is not attenuated by imatinib. It might be interesting to see whether VEGF-B mediated Abl kinase activation contributes to the impaired insulin-mediated vasodilation observed in this study.

## Strengths & limitations

The combination of in vivo and in vitro data in this study provide insight into the anti-diabetic effect of imatinib and the underlying physiological mechanisms. We have been able to show that imatinib preserves endothelial function by direct examination of the effect of insulin on resistance artery diameter using intravital microscopy. We have not only focused on one possible physiological mechanism that could explain the anti-diabetic effect of imatinib, but have also shown that altered lipid metabolism does not explain this effect, even though imatinib did prevent weight-gain. The in vitro data nicely supplement the in vivo data here by showing that, while imatinib did affect VEGF-B mediated signaling, it did not affect fatty-acid uptake. By studying both the vascular effects of imatinib and the effects on lipid metabolism we have been able to narrow down why imatinib has anti-diabetic effects. Another strength of this study is our use of a model for early diet-induced weight gain and insulin resistance, because endothelial dysfunction is seen in the early stages of disease development in type 2 diabetes. The fact that we show here that imatinib preserves an important endothelial function that is part of healthy vascular insulin sensitivity (insulin-mediated vasodilation) in a model for early diet-induced insulin resistance shows that imatinib affects a process that is a cornerstone of type 2 diabetes development. In future research it would also be very interesting to study the direct effects of imatinib on insulin mediated vasodilation ex-vivo to see whether our current results were a direct or indirect effect of imatinib. Furthermore, it would be of value to study the effect of imatinib on the later stages of the type 2 diabetes, using db/db mice for instance, especially to study the effect on metabolic insulin sensitivity. This was beyond the scope of the current study.

This study also has its limitations. Regarding the in vivo part of the study, we firstly used a supra-physiological insulin infusion rate in our hyperinsulinemic-euglycemic clamp. This might have affected whole-body glucose uptake since, at supra-physiological concentrations, insulin also binds to the more abundant IGF-1 receptor [47]. Secondly, we treated with imatinib for a relatively short period, only two weeks. In other studies that did show the effect of imatinib on triglycerides mice were treated longer. The effects of imatinib on lipid metabolism might need longer treatment to show than its effects on insulin-mediated vasodilation. Thirdly, we did not have any data on imatinib concentration in plasma and tissue, fat distribution or amount of fat tissue in the mice. These data would have helped us understand where imatinib works and give us more insight into the effects on lipid metabolism.

## Conclusion

In conclusion, imatinib prevents weight gain in WD-fed mice and preserves insulin-mediated vasodilation, independent of triglyceride content and fatty acid transport. Instead, the

protective effect of imatinib on vascular insulin sensitivity, as shown by the significant rescue of insulin-mediated vasodilation of resistance arteries in vivo by imatinib, likely contributes to its anti-diabetic effects. Hence, this study has provided new insights into the possible mechanisms responsible for the anti-diabetic effects of imatinib and will lead to focused research on a new therapeutic option for type 2 diabetes and its complications.

## Supporting information

**S1 Fig. For descriptions see Figs 4 and 5 in main article text.**
(TIF)

## Acknowledgments

The authors thank Zeineb Gam and Alinda Schimmel for excellent technical support.

## Author Contributions

**Conceptualization:** Alexander H. Turaihi, Jurjan Aman, Etto C. Eringa.

**Data curation:** Camiel V. J. Box, Amandeep K. Sandhu, Jurjan Aman, Etto C. Eringa.

**Formal analysis:** Camiel V. J. Box, Amandeep K. Sandhu, Jurjan Aman, Etto C. Eringa.

**Investigation:** Camiel V. J. Box, Amandeep K. Sandhu, Alexander H. Turaihi, Pan Xiaoké, Geesje Dallinga-Thie.

**Methodology:** Camiel V. J. Box, Amandeep K. Sandhu, Alexander H. Turaihi, Pan Xiaoké, Geesje Dallinga-Thie, Jurjan Aman, Etto C. Eringa.

**Supervision:** Jurjan Aman, Etto C. Eringa.

**Validation:** Jurjan Aman, Etto C. Eringa.

**Visualization:** Camiel V. J. Box, Amandeep K. Sandhu.

**Writing – original draft:** Camiel V. J. Box, Amandeep K. Sandhu, Jurjan Aman, Etto C. Eringa.

**Writing – review & editing:** Camiel V. J. Box, Amandeep K. Sandhu, Jurjan Aman, Etto C. Eringa.

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
