## [Decision Letter · Decision Letter 0]

3 Mar 2020

PONE-D-19-35443

Effect of imatinib on endothelial dysfunction in early weight gain: vascular insulin sensitivity and free fatty acid transport in vivo and in vitro

PLOS ONE

Dear Mr Box,

Thank you for submitting your manuscript to PLOS ONE. After careful consideration, we feel that it has merit but does not fully meet PLOS ONE’s publication criteria as it currently stands. Therefore, we invite you to submit a revised version of the manuscript that addresses the points raised during the review process.

We would appreciate receiving your revised manuscript by Apr 17 2020 11:59PM. To enhance the reproducibility of your results, we recommend that if applicable you deposit your laboratory protocols in protocols.io, where a protocol can be assigned its own identifier (DOI) such that it can be cited independently in the future. For instructions see: http://journals.plos.org/plosone/s/submission-guidelines#loc-laboratory-protocols

We look forward to receiving your revised manuscript.

Kind regards,

Christopher Torrens

Academic Editor

PLOS ONE

Journal Requirements:

Reviewers' comments:

Reviewer's Responses to Questions

**Comments to the Author**

1. Is the manuscript technically sound, and do the data support the conclusions?

Reviewer #1: Partly

Reviewer #2: Partly

2. Has the statistical analysis been performed appropriately and rigorously? 

Reviewer #1: Yes

Reviewer #2: Yes

3. Have the authors made all data underlying the findings in their manuscript fully available?

Reviewer #1: Yes

Reviewer #2: Yes

4. Is the manuscript presented in an intelligible fashion and written in standard English?

Reviewer #1: Yes

Reviewer #2: Yes

5. Review Comments to the Author

Reviewer #1: This manuscript describes a potentially interesting mechanism by which VEGF-B, through Arg, may have positive effects on fatty acid transport and metabolism and that imatinib inhibits some of these. It is difficult to assess if the final conclusions are valid due to the study design and limitations. Two major concerns are raised along with a few minor comments.

If body weight is not equivalent, a change in insulin sensitivity and glucose levels is expected. A beneficial finding is thus moot. It seems that the authors are celebrating this finding as positive, when really the mouse exhibits a lipodystrophy-like phenotype. Thus, despite being lower in mass, glucose levels are equivalent (close to being higher even on chow) and insulin sensitivity is barely different. What are the actual masses of these mice (not % change)? Body composition? That muscle takes on equivalent lipid, despite leaner mice, is of metabolic concern.

In vitro, wouldn't inhibited phosphorylation of downstream VEGFR2 targets be expected as imatinib is a broad-spectrum tyrosine kinase inhibitor? Data on pErk or pCrkl is therefore not very interesting. Why would the authors expect differently? The mechanism of imatinib action must be described in the intro and explained relative to the results.

The most significant finding is the inability of imatinib to impact VEGF-B mediated FA uptake. This is very interesting and the authors should focus on discussing this mechanism and how it fits into past VEGF-B FA transporter work, VEGFR2 signaling, and the 'controversy' of whether/how VEGFB does this. If imatinib inhibits tyrosine kinases, if there some other pathway?

Minor: Several of the figure panels should probably be combined. Currently 1+2+3, though this may change if the authors provide body composition or muscle pAkt.

Minor: the physiology of vessel dilation is commendable, but multiple mechanisms could be at play (1) if truly more insulin sensitive, this result is expected (2) discussion of the VEGFR2-nitric oxide axis?

Minor: use of standard deviations would better illustrate why the results that are close but not significant are not actually signification.

Minor: headings in the conclusion are required?

Minor: line 379. to say that it is protective is not valid if no tissue pAkt response or other factors are examined? permeability?

Minor: line 434. Only uptake of FA is shown, not permeability.

Minor: no discussion of Robciuc 2016 on VEGFB mice being leaner is included.

Minor: final limitations: the authors mentions several but fail to address some of the criticisms raised by this reviewer that should seem obivous?

Reviewer #2: General comments:

Inconsistent format appears in the manuscript. For example: Page 8 line 206: two-way ANOVA and Page 9 line 235: 2-way ANOVA.

Most of the content in the introduction is about type 2 diabetes, but the research related to this aspect is relatively insufficient. Whether use DB mouse model for further research to investigate the effects of imatinib on metabolic insulin sensitivity?

Specific comments:

1. Introduction: The ideas presented (Since VEGF-A and VEGF B stimulate the same receptor，Page 4 lines 92 through 93) are not rigorous. VEGF-A is the prototypical member of the family and a strong angiogenic inducer, predominately by signaling through VEGFR2. VEGF-B binds specifically to VEGFR1 and to the common co-receptor neuropilin-1 (NRP1) to regulate lipid metabolism.

2. Methods: Page 4 line 111: Most dietary lipids are long-chain FAs (LCFAs; >12 carbons), whether the product of C1-BODIPY®500/510 C12 is your best choice to study endothelial FA transport in vitro?

3. Results:

a. An expression error of ImatWD mice in saline treated mice appear Page 9 line 227. Line 234: Weight-gaining effect was rescued by imatinib treatment (P=0.006), why express P-values only one *?

b. Figure 2 need more evidence to examine the effects of imatinib on metabolic insulin sensitivity (for example: ITT). Why are there decreasing trends of the fasting blood glucose level in WD mice comparing with chow diets with or without imatinib treatment?

c. Figure 4: the same expression of P-values* in figure 4. Page 11 line 284: there is an expression error of (P>0.9).

d. Figure 5: Writing errors of VEGF-B (100 ng/ml) in figure 5 A-D. Whether VEGFB and imatinib treat endothelial cells together can better explain imatinib affects the VEGF-B mediated phosphorylation of CrkL.

a. Figure 7: More experiments is needed to improve VEGF-B stimulation caused a increase in the uptake of fatty acids corresponding with increased phosphorylation of Arg substrate CrkL at Tyr207. There need an additional treated group of VEGFB、VEGFB antibody and imatinib to further improve the effect of imatinib on VEGFB mediated endothelial FA uptake.

4. Discussion:

1. Page 14 line 382: The available data do not support this view.

2. As mentioned in the discussion (Page 16 lines 414 through 417), the treatment time in mouse model may be too short and could be extended.

3. Insufficient data cannot explain the opinion that Abl kinases mediate VEGF-B signaling without affecting VEGFB-mediated uptake of fatty acids.

6. PLOS authors have the option to publish the peer review history of their article (what does this mean?). If published, this will include your full peer review and any attached files.

Reviewer #1: No

Reviewer #2: No

---

## [Author Response · Author response to Decision Letter 0]

17 Sep 2020

The comments from the reviewers have been very helpful in revising and improving the manuscript. We would therefore like to thank them for taking the time to read our manuscript and provide their insights. We have responded to every comment in the attached rebuttal letter and revised the manuscript accordingly. Also, we have performed additional experiments to supplement the data. In muscle, the added data on expression of VEGFB, VEGFR1, IRS1, and Glut4 provide mechanistic insight into the relationship between lipid and glucose uptake in muscle during early weight gain.

---

## [Decision Letter · Decision Letter 1]

3 Feb 2021

PONE-D-19-35443R1

Effects of imatinib on vascular insulin sensitivity and free fatty acid transport in early weight gain

PLOS ONE

Dear Dr. Box,

Thank you for submitting your manuscript to PLOS ONE. After careful consideration, we feel that it has merit but does not fully meet PLOS ONE’s publication criteria as it currently stands. Therefore, we invite you to submit a revised version of the manuscript that addresses the points still raised by reviewer 1.

We look forward to receiving your revised manuscript.

Kind regards,

Michael Bader

Academic Editor

PLOS ONE

Reviewers' comments:

Reviewer's Responses to Questions

**Comments to the Author**

1. If the authors have adequately addressed your comments raised in a previous round of review and you feel that this manuscript is now acceptable for publication, you may indicate that here to bypass the “Comments to the Author” section, enter your conflict of interest statement in the “Confidential to Editor” section, and submit your "Accept" recommendation.

Reviewer #1: (No Response)

2. Is the manuscript technically sound, and do the data support the conclusions?

Reviewer #1: Partly

3. Has the statistical analysis been performed appropriately and rigorously? 

Reviewer #1: Yes

4. Have the authors made all data underlying the findings in their manuscript fully available?

Reviewer #1: Yes

5. Is the manuscript presented in an intelligible fashion and written in standard English?

Reviewer #1: Yes

6. Review Comments to the Author

Reviewer #1: The authors have provided some re-writing and edits to help improve the manuscript, but have failed to addressed several concerns due to these requiring further experimentation. These concerns could be addressed by discussing them, rather than merely arguing back to the reviewer that they are unnecessary. Much of this stems from the confusion of the authors trying to integrate imatinib and VEGF-B signaling together, yet not successfully doing either well in of itself. The concluding paragraph of the introduction states that "aim of our study was to investigate the effect of imatinib on endothelial fatty acid uptake and on vascular insulin sensitivity through insulin-mediated vasodilation of resistance arteries". By bringing VEGFB into this, and the conflicting reports of its role in FA uptake (Dijkstra 2014)(or even its intracellular signaling), the message is muddy.

Perhaps VEGFB in the intro and throughout could be downplayed and only introduced as a means to elicit a change in FA uptake? The extensive discussion of this factor makes the reviewer read thinking that the purpose of imatinib is to counter VEGFB throughout... but I do not think that is the authors' purpose?

"aim of our study was to investigate the effect of imatinib... on vascular insulin sensitivity through insulin-mediated vasodilation of resistance arteries" This study is commendable, but it is not clear that the experiment is achieving this aim. The aim would be on isolated vessels independent of the mouse's metabolic state, but these are in situ in mice with preserved insulin sensitivity, so the result is less interesting than a direct competition. Would imatinib in the bath of an isolated vessel from an insulin-resistant mouse, or if in situ, direct application to the vessel region examined, elicit the same response? Else this is just further confirmation, like the other metabolic data, that, yes, imatinib-treated animals are healthier. A minor point along those lines is that the authors should not use 'improve' or 'increase' but rather the vasodilatory response is 'preserved' in mice receiving imatinib since this is not a therapeutic application.

7. PLOS authors have the option to publish the peer review history of their article (what does this mean?). If published, this will include your full peer review and any attached files.

Reviewer #1: No

---

## [Author Response · Author response to Decision Letter 1]

2 Apr 2021

Response to reviewer:

Thank you for your time and careful review of our paper. We will answer to each of your points below:

Comment #1: “The authors have provided some re-writing and edits to help improve the manuscript, but have failed to addressed several concerns due to these requiring further experimentation. These concerns could be addressed by discussing them, rather than merely arguing back to the reviewer that they are unnecessary. Much of this stems from the confusion of the authors trying to integrate imatinib and VEGF-B signaling together, yet not successfully doing either well in of itself. The concluding paragraph of the introduction states that "aim of our study was to investigate the effect of imatinib on endothelial fatty acid uptake and on vascular insulin sensitivity through insulin-mediated vasodilation of resistance arteries". By bringing VEGFB into this, and the conflicting reports of its role in FA uptake (Dijkstra 2014)(or even its intracellular signaling), the message is muddy.

Perhaps VEGFB in the intro and throughout could be downplayed and only introduced as a means to elicit a change in FA uptake? The extensive discussion of this factor makes the reviewer read thinking that the purpose of imatinib is to counter VEGFB throughout... but I do not think that is the authors' purpose?”

Reply: The reviewer’s point is well taken, the relationship between insulin’s effects and VEGF-B was indeed not very clear in the introduction. In response we have explained the relationship between cellular fatty acid accumulation and insulin-induced vasodilation in the Introduction (p.3, ll.67-72) as follows:

“Increased levels of FFAs directly impair insulin mediated vasodilation46 and trigger low-grade inflammation, which enhances endothelial permeability [23]. One of the potential regulators of fatty acids uptake is vascular endothelial growth factor-B (VEGF-B), which has been proposed to stimulate transendothelial transport (TET) of lipids [5–7]. Although regulation of fatty acid transport and blood flow are key features of endothelial dysfunction in type 2 diabetes, few therapeutic agents improve these processes[6].”

Ref. 21 shows a direct interaction of fatty acids within endothelial cells and insulin’s effects in muscle resistance arteries, linking the two parts of our study.

We agree that the part of the discussion on VEGF-B was a bit extensive and have condensed it following the reviewer’s suggestion. 

Comment #2: "aim of our study was to investigate the effect of imatinib... on vascular insulin sensitivity through insulin-mediated vasodilation of resistance arteries" This study is commendable, but it is not clear that the experiment is achieving this aim. The aim would be on isolated vessels independent of the mouse's metabolic state, but these are in situ in mice with preserved insulin sensitivity, so the result is less interesting than a direct competition. Would imatinib in the bath of an isolated vessel from an insulin-resistant mouse, or if in situ, direct application to the vessel region examined, elicit the same response? Else this is just further confirmation, like the other metabolic data, that, yes, imatinib-treated animals are healthier. A minor point along those lines is that the authors should not use 'improve' or 'increase' but rather the vasodilatory response is 'preserved' in mice receiving imatinib since this is not a therapeutic application.”

Reply: In response to the reviewers observation we have made revisions to the article where it describes this particular aim and experiment, in the introduction, results, and discussion section. We indeed should not have stated that our study directly aims to show an effect of imatinib on vascular insulin sensitivity through insulin mediated vasodilation of resistance arteries, instead our study shows how imatinib preserves insulin-mediated vasodilation of resistance arteries in vivo in mice fed a Western diet. While insulin-mediated vasodilation of resistance arteries is a defining characteristic of normal vascular effects of insulin, it is certainly not the only one and therefore the reviewer is correct in stating that our aim should have been more specific. As the reviewer also rightfully states it would indeed be of added value to investigate whether imatinib elicits this effect directly in an ex-vivo setting to see whether this effect of imatinib in vivo was directly or through some other pathway that is not present in an ex-vivo setting. We have included this suggestion in our discussion section as an interesting part of follow-up research, but answering this question would require a new study on interaction of imatinib with fatty acid impairment of insulin-induced vasodilation of isolated muscle resistance arteries. Thirdly we have, per suggestion of the reviewer, corrected the sentences that stated that the vasodilatory response wat improved and now used the word “preserved” which is indeed more apt.

---

## [Editor Report · Decision Letter 2]

7 Apr 2021

Effects of imatinib on vascular insulin sensitivity and free fatty acid transport in early weight gain

PONE-D-19-35443R2

Dear Dr. Box,

We’re pleased to inform you that your manuscript has been judged scientifically suitable for publication and will be formally accepted for publication once it meets all outstanding technical requirements.

Kind regards,

Michael Bader

Academic Editor

PLOS ONE
---

## [Editor Report · Acceptance letter]

24 Jun 2021

PONE-D-19-35443R2 

Effects of imatinib on vascular insulin sensitivity and free fatty acid transport in early weight gain 

Dear Dr. Box:

I'm pleased to inform you that your manuscript has been deemed suitable for publication in PLOS ONE. Congratulations! Your manuscript is now with our production department. 

Kind regards, 

on behalf of

Prof. Michael Bader 

Academic Editor

PLOS ONE